# Primary Effusion Lymphoma: A Clinicopathological Study of 70 Cases

**DOI:** 10.3390/cancers13040878

**Published:** 2021-02-19

**Authors:** Zhihong Hu, Zenggang Pan, Weina Chen, Yang Shi, Wei Wang, Ji Yuan, Endi Wang, Shanxiang Zhang, Habibe Kurt, Brenda Mai, Xiaohui Zhang, Hui Liu, Adan A. Rios, Hilary Y. Ma, Nghia D. Nguyen, L. Jeffrey Medeiros, Shimin Hu

**Affiliations:** 1Department of Pathology and Laboratory Medicine, The University of Texas Health Science Center at Houston, Houston, TX 77030, USA; Brenda.Mai@uth.tmc.edu (B.M.); Nghia.D.Nguyen@uth.tmc.edu (N.D.N.); 2Department of Pathology, Yale University School of Medicine, New Haven, CT 06510, USA; zenggang.pan@yale.edu; 3Department of Pathology, The University of Texas Southwestern Medical Center, Dallas, TX 75390, USA; Weina.Chen@UTSouthwestern.edu; 4Department of Pathology, Montefiore Medical Center/Albert Einstein College of Medicine, Bronx, NY 10461, USA; yash@montefiore.org; 5Department of Hematopathology, The University of Texas MD Anderson Cancer Center, Houston, TX 77030, USA; wwang13@mdanderson.org (W.W.); hliu@xzhmu.edu.cn (H.L.); ljmedeiros@mdanderson.org (L.J.M.); 6Department of Pathology and Microbiology, University of Nebraska Medical Center, Omaha, NE 68198, USA; Yuan.Ji@mayo.edu; 7Department of Pathology, Duke University Medical Center, Durham, NC 27710, USA; endi.wang@duke.edu; 8Department of Pathology, Indiana University School of Medicine, Indianapolis, IN 46202, USA; sz5@iupui.edu; 9Department of Pathology and Laboratory Medicine, Rhode Island Hospital and Warren Alpert Medical School of Brown University, Providence, RI 02912, USA; HKurt@Lifespan.org; 10Department of Pathology, H. Lee Moffitt Cancer Center and Research Institute, Tampa, FL 33612, USA; Xiaohui.Zhang@moffitt.org; 11Oncology Division, Department of Internal Medicine, The University of Texas Health Science Center/McGovern Medical School at Houston, Houston, TX 77030, USA; Adan.Rios@uth.tmc.edu; 12Department of General Oncology, The University of Texas MD Anderson Cancer Center, Houston, TX 77030, USA; HYMa@mdanderson.org

**Keywords:** primary effusion lymphoma, extracavitary variant, HHV8, HIV, EBV, Kaposi sarcoma, highly active antiretroviral therapy

## Abstract

**Simple Summary:**

Primary effusion lymphoma (PEL) is a rare HHV8 driven large B-cell lymphoma. It is often associated with HIV infection and seldom occurs in HIV-negative immunocompromised patients. Patients with PEL usually present with effusion only, but occasionally with an extracavitary mass, or both. This retrospective study aimed to better characterize the clinicopathological features of PEL by comparing effusion-only PEL versus the extracavitary-only PEL and HIV-positive versus HIV-negative cases in a large cohort of 70 patients. All 70 (100%) cases were positive for HHV8. Fifty-six (80%) patients had HIV infection. Patients presenting with effusion only versus extracavitary disease were associated with different clinicopathologic features. After a median follow-up time of 40 months (range 0–96), 26 of 52 (50%) patients with clinical follow-up died, and the median survival was 42.5 months. PEL is an aggressive lymphoma with a poor prognosis, regardless of extracavitary presentation or HIV status.

**Abstract:**

Primary effusion lymphoma (PEL) is a rare type of large B-cell lymphoma associated with human herpesvirus 8 (HHV8) infection. Patients with PEL usually present with an effusion, but occasionally with an extracavitary mass. In this study, we reported a cohort of 70 patients with PEL: 67 men and 3 women with a median age of 46 years (range 26–91). Of these, 56 (80%) patients had human immunodeficiency virus (HIV) infection, eight were HIV-negative, and six had unknown HIV status. Nineteen (27%) patients had Kaposi sarcoma. Thirty-five (50%) patients presented with effusion only, 27 (39%) had an extracavitary mass or masses only, and eight (11%) had both effusion and extracavitary disease. The lymphoma cells showed plasmablastic, immunoblastic, or anaplastic morphology. All 70 (100%) cases were positive for HHV8. Compared with effusion-only PEL, patients with extracavitary-only PEL were younger (median age, 42 vs. 52 years, *p* = 0.001), more likely to be HIV-positive (88.9% vs. 68.6%, *p* = 0.06) and EBV-positive (76.9% vs. 51.9%, *p* = 0.06), and less often positive for CD45 (69.2% vs. 96.2%, *p* = 0.01), EMA (26.7% vs. 100%, *p* = 0.0005), and CD30 (60% vs. 81.5%, *p* = 0.09). Of 52 (50%) patients with clinical follow-up, 26 died after a median follow-up time of 40.0 months (range 0–96), and the median overall survival was 42.5 months. The median OS for patients with effusion-only and with extracavitary-only PEL were 30.0 and 37.9 months, respectively (*p* = 0.34), and patients with extracavitary-only PEL had a lower mortality rate at the time of last follow-up (35% vs. 61.5%, *p* = 0.07). The median OS for HIV-positive and HIV-negative patients were 42.5 and 6.8 months, respectively (*p* = 0.57), and they had a similar mortality rate of 50% at last follow-up. In conclusion, patients presenting with effusion-only versus extracavitary-only disease are associated with different clinicopathologic features. PEL is an aggressive lymphoma with a poor prognosis, regardless of extracavitary presentation or HIV status.

## 1. Introduction

In the 2016 revised World Health Organization (WHO) classification of tumors of hematopoietic and lymphoid tissues, primary effusion lymphoma (PEL) is defined as an aggressive type of large B-cell lymphoma that is universally associated with human herpesvirus 8 (HHV8) infection [1]. PEL was first identified to be a HHV8-driven body-cavity based lymphoma in human immunodeficiency virus (HIV)-positive patients in 1995 [2]. It was subsequently included in the 2001 WHO classification of lymphoid neoplasms [3]. PEL usually presents as a serous effusion without a detectable tumor mass, but occasionally patients may present with an extracavitary mass [4]. Most patients develop PEL in the context of HIV infection, but PEL also can occur in HIV-negative patients. 

HHV8, also called Kaposi sarcoma-associated herpes virus, was first identified in patients with acquired immunodeficiency syndrome [5]. Besides PEL and Kaposi sarcoma, HHV8 is linked to several other diseases including HIV-associated multicentric Castleman disease, HHV8-positive diffuse large B-cell lymphoma not otherwise specified, HHV8-positive germinotropic lymphoproliferative disorder, and HHV8 inflammatory cytokine syndrome [6]. Viral interleukin-6 (vIL-6) is involved in the pathogenesis of HHV8-associated diseases. PEL arises from HHV8 infected B cells and vIL-6 promotes tumor cell survival and proliferation [7,8,9,10].

Others have reported the clinicopathologic features of patients with PEL. The extracavitary variant of PEL has been described, and there appears some differences between effusion-only and extracavitary PEL [11,12]. However, to date no large cohort of PEL cases has been assessed systematically for potential differences between PEL patients who present only with an effusion versus PEL patients who present with extracavitary disease. Furthermore, few studies have compared PEL in patients with HIV infection versus PEL in HIV-negative patients. In this study, we aimed to better characterize this entity by specifically comparing effusion-only versus extracavitary-only disease and HIV-positive versus HIV-negative cases in a large cohort of 70 patients with PEL. 

## 2. Materials and Methods

### 2.1. Patients 

Cases of PEL diagnosed from 1 January 2004 through 30 June 2019 in 10 different institutions were reviewed. The institutions included The University of Texas MD Anderson Cancer Center, The University of Texas Health Science Center at Houston, Yale University School of Medicine, The University of Texas Southwestern Medical Center, Montefiore Medical Center/Albert Einstein College of Medicine, University of Nebraska Medical Center, Duke University Medical Center, Indiana University Medical Center, Rhode Island Hospital/Brown University, and H. Lee Moffitt Cancer Center and Research Institute. All cases included in this study met the diagnostic criteria of the revised 2016 WHO classification for the diagnosis of PEL. Clinical parameters, laboratory data, treatment, and outcome information were collected from the medical record systems. The study was approved by the Institutional Review Boards of The University of Texas Health Science Center at Houston and participating institutions.

### 2.2. Flow Cytometric Immunophenotyping

Flow cytometric immunophenotypic studies were performed on effusion fluid, lymph node, extranodal mass, and/or bone marrow aspirate specimens using multicolor analysis. The antibody panels varied in the different institutions and evolved significantly during the study interval. Overall, the flow panels included antibodies for CD2, CD3, CD4, CD5, CD7, CD8, CD10, CD19, CD20, CD22, CD30, CD38, CD45, CD138, kappa and lambda (surface and cytoplasmic) immunoglobulin light chains (Becton-Dickinson, Biosciences, San Jose, CA, USA). Data were analyzed using FCS Express (De novo Software, Glendale, CA). Antigen expression was assessed on lymphoma cells with antibody isotype negative controls and was considered positive when at least 20% of lymphoma cells had the expression, an arbitrary cutoff used to define a significant expression in most of the biomarker studies. 

### 2.3. Histological Examination and Immunohistochemistry Analysis

Diff-Quick and Papanicolaou stains were performed on cytospins of effusion fluid specimens. Hematoxylin-eosin stained slides of lymph node, extranodal mass, fluid cell block, and bone marrow core biopsy specimens were prepared and reviewed. Wright Giemsa stain was performed on peripheral blood and bone marrow aspirate smears. Immunohistochemical analysis was performed using on 4-μM-thick histologic sections prepared from formalin-fixed paraffin-embedded tissue biopsy or cell block specimens. The antibodies assessed were specific for CD3, CD20, CD30, CD45, CD79a, CD138, ALK-1, PAX-5, MUM-1, EMA, and HHV-8. In situ hybridization analysis was also performed on subsets of patients for EBV-encoded small RNAs (EBER) as well as immunoglobulin kappa and lambda light chains. 

### 2.4. Fluorescence In Situ Hybridization (FISH) Studies

FISH analysis was performed on 4-μM-thick sections of formalin-fixed paraffin-embedded tissue blocks from lymph node and extranodal mass, and/or cell block of effusion fluid specimens according to the standard protocols. The LSI *MYC* dual color break-apart probe kit (Abbott Laboratories, Abbott Park, IL, USA) was utilized in this study. A total of 200 interphase nuclei were assessed.

### 2.5. Statistical Analysis

Data for qualitative variables such as gender and complete blood count abnormalities including anemia, leukopenia and thrombocytopenia were expressed as the number of specified patients to the total patients. Data for quantitative variables such as age and follow-up time were defined as median and range. Chi-square test was used to compare the significance of difference of categorical variables. Fisher’s exact test was performed when there was a cell with a value of “0”. Log-rank test was performed to compare the overall survival (OS) of different subgroups by using the statistical software of GraphPad prism 8 (GraphPad Software, San Diego, CA, USA). Statistical significance was considered when the *p* value was <0.05.

## 3. Results

### 3.1. Clinical Features

The study cohort included 70 patients with PEL: 67 men and 3 women, with a median age of 46 years (range, 26–91) at the time of diagnosis. Fifty-six (80%) patients were HIV-positive, eight (11.4%) were HIV-negative including two heart transplant recipients, and six (8.6%) had unknown HIV status (Table 1). The HIV-positive patients had a median age of 45 years (range 26–77) and the HIV-negative patients had a median age of 75 years (range 51–88) (*p* = 0.005). Twenty-six HIV-positive patients were known to have received highly active anti-retroviral therapy (HAART) before their diagnosis of PEL, and 12 patients were noncompliant. Whether other patients received HAART was not known. Thirty-five HIV-positive patients had available CD4+ T-cell count information at diagnosis of PEL, and 27 had CD4 count < 200/μL. Twenty-nine patients had available CD4/CD8 ratio with a median of 0.13 (range, 0.04–3.91). The median viral load at diagnosis of PEL in 22 patients with available information was 138,518 copies/μL (range, <20–406,000). 

Thirty-five (50%) patients (24 HIV-positive, seven HIV-negative, and four unknown) presented only with an effusion (24 pleural, eight pericardial, and seven peritoneal), including four patients with multiple effusions (two with both pleural and pericardial; two with both pleural and peritoneal). Twenty-seven (39%) patients (24 HIV-positive, one HIV-negative, and two unknown HIV status) had only an extracavitary mass or masses (16 nodal, eight gastrointestinal tract, two perirenal, two skin/buccal mucosal, one cardiac atrial, one central nervous system (CNS), and one pelvic), including four patients with multiple masses (three with gastrointestinal tract and nodal; one with CNS and nodal). The remaining eight (11%) patients (all HIV-positive) had both an effusion (four pleural, three peritoneal, and one pericardial) and extracavitary disease (three nodal, one duodenal, one perirenal, one inguinal and cutaneous, one cardiac atrial, and one pleural). 

Nineteen (27%) patients had Kaposi sarcoma, including 17 known to be HIV-positive and two with unknown HIV status. None of eight HIV-negative patients had Kaposi sarcoma. Of the 19 patients with Kaposi sarcoma, nine had effusion only, seven had extracavitary disease only, and three had both effusion and extracavitary disease. 

### 3.2. Morphology, Immunophenotype, and Cytogenetic Findings

In effusion specimens, the lymphoma cells were large and discohesive, and displayed round or irregular nuclear contours with variably prominent nucleoli and moderately abundant agranular basophilic cytoplasm. Occasionally, the lymphoma cells had sparse cytoplasmic vacuoles. These cells showed variable morphology, resembling plasmablasts or immunoblasts, or with anaplastic features (Figure 1). Occasional atypical mitotic figures were seen. Extracavitary mass specimens were composed of cells with similar cytomorphologic features. In general, PEL cells in the extracavitary masses showed two different infiltrative patterns: sinusoidal and diffuse. The morphology and immunophenotype of representative cases of effusion-only and extracavitary-only PEL are shown in Figure 2 and Figure 3. 

Histologic examination of Kaposi sarcoma in these patients showed a spindle cell proliferation associated with many extravasated erythrocytes (Figure 4A). The spindle cells were positive for HHV8 (Figure 4A inset). Figure 4 showed one representative case of PEL in a patient with a history of Kaposi sarcoma. The PEL cells in this case had a sinusoidal pattern of infiltration (Figure 4B).

Immunohistochemical analysis of all 70 PEL cases showed that the lymphoma cells were positive for HHV8. Combining the results of flow cytometry and immunohistochemistry analysis, the PEL cells were positive for the following markers: CD38 32/34 (94.1%), MUM1/IRF4 31/36 (86.1%), CD45 50/59 (84.7%), CD30 44/60 (73.3%), CD43 8/14 (57.1%), EMA 16/28 (57.1%), CD138 30/58 (51.7%), monotypic kappa or lambda 15/37 (40.5%) (seven monotypic for kappa and eight monotypic for lambda; the remaining cases lacked surface immunoglobulin light chain expression), CD3 14/65 (21.5%), CD79a 5/37 (13.5%), weak CD20 5/66 (7.6%), and rare PAX-5-positive cells 1/26 (3.8%) (Table 2). The PEL cells were negative for CD10 (*n* = 10), CD19 (*n* = 20), and ALK-1 (*n* = 34). The Ki67 proliferation rate was ≥70% in 25/26 (96.1%) cases, with a median of 90% (range, 50–100%). In situ hybridization showed that EBER was positive in 39/60 (65%) cases. FISH analysis was performed on five cases and showed an extra copy of *MYC* in 4 (80%) cases. There was no evidence of *MYC* rearrangement.

### 3.3. Staging, Treatment, and Outcome

Twenty-three patients underwent bone marrow staging, and no PEL involvement was identified in the bone marrow. Twenty-six patients had lymphoma treatment information available including 13 treated with dose-adjusted etoposide, doxorubicin, cyclophosphamide, vincristine, and prednisone (DA-EPOCH), eight with cyclophosphamide, doxorubicin, vincristine, and prednisone (CHOP), and five with other chemotherapy regimens (2 with doxil/paclitaxel, 1 with rituximab, ifofamide, carboplastin, and etoposide, 1 with cyclophosphamide, vincristine, and prednisone, and 1 with Bendamustine). Of 52 patients with known survival status, 50 had exact last follow-up time available; 25 (50%) died after a median follow-up time of 40.0 months (range 0–96). The median overall survival (OS) was 42.5 months. In the remaining two patients (one alive; one dead), the exact last follow-up time was not available.

### 3.4. Clinicopathologic and Prognostic Difference between Patients with Effusion-Only versus with Extracavitary-Only PEL

The clinical and pathologic variables between patients with effusion-only PEL versus patients with extracavitary-only PEL are summarized in Table 2 and Table 3. Patients with extracavitary-only PEL were younger than patients with effusion-only PEL (median age, 42 vs. 52 years, *p* = 0.001) and more likely HIV-positive (88.9% vs. 68.6%, *p* = 0.06) (Table 3). The lymphoma cells in extracavitary-only PEL were less often positive for CD45 (69.2% vs. 96.2%, *p* = 0.01), CD30 (60.0% vs. 81.5%, *p* = 0.09), and EMA (26.7% vs. 100%, *p* = 0.0005), and were more often positive for EBER (76.9% vs. 51.9%, *p* = 0.06) and CD3 expression (33.3% vs. 13.3%, *p* = 0.07) (Table 2). There were no differences in the expression of other phenotypic markers, nor any differences in laboratory findings or treatment between these two subgroups. 

With a median follow-up time of 40.0 months (range, 0–96), the median OS for patients with effusion-only PEL and patients with extracavitary-only PEL were 30.0 and 37.9 months, respectively (*p* = 0.34) (Figure 5A). Patients with extracavitary-only PEL had a lower mortality rate at the time of last follow-up (35% vs. 61.5%, *p* = 0.07).

### 3.5. Clinicopathologic and Prognostic Differences between PEL Patients with HIV versus without HIV Infection

Patients with HIV infection had a median age of 45 years (range 26–77) versus 75 years (range 51–88) for HIV-negative patients (*p* = 0.005) (Table 4). Compared with HIV-positive patients, HIV-negative patients tended to have a lower frequency of extracavitary disease (*p* = 0.10). Seven of eight (87.5%) HIV-negative patients had effusion-only PEL and one had extracavitary disease only. In contrast, 24 of 56 (42.9%) HIV-positive patients had effusion only, 24 (42.9%) had extracavitary disease only, and eight (14.3%) had both effusion and extracavitary disease. In addition, HIV-negative patients had a lower frequency of EBER positivity (37.5% vs. 75.0%, *p* = 0.03). The median OS for HIV-positive and HIV-negative patients were 42.5 and 6.8 months, respectively (*p* = 0.57) (Figure 5B). HIV-positive and HIV-negative patients had a similar mortality rate of 50% at last follow-up.

## 4. Discussion

In this large cohort of PEL cases, we compared the clinicopathologic and immunophenotypic differences between effusion-only versus extracavitary-only PEL, and we identified some differences. Patients with extracavitary-only PEL tended to be younger and were more likely to be HIV-positive and EBV-positive. The lymphoma cells of extracavitary-only PEL were also less often positive for CD45, CD30, and EMA, but more often positive for CD3 compared with effusion-only PEL. The findings of frequent lack of CD45 expression in extracavitary-only PEL are consistent with previous reports [4]. However, it is unclear whether these differences reflect variations in tumor biology or the influence of the tumor microenvironment. 

HIV infection is an important factor involved in the pathogenesis of PEL [13,14], and most patients (80%) in this cohort were HIV-positive. These patients had a median age of 45 years. In contrast, the median age of the HIV-negative group was 75 years. In HIV-positive patients, HHV8 infects human B cells (and other cells) and encodes viral oncogenes such as vIL-6, viral homolog of the Fas-associated death domain-like IL-1-β-converting enzyme inhibitory protein; viral dysregulation of human cells is likely to play a role in the development of PEL [15,16,17,18]. Chronic antigen stimulation and overproduction of cytokines including cellular IL-6 and IL-10 in the HIV-positive patients also plays a potential role in PEL pathogenesis, similar to what occurs in other non-Hodgkin B cell lymphomas [15,16,19,20,21]. Two patients in the HIV-negative group had received a heart transplant, implicating iatrogenic immunosuppression in their pathogenesis. The older age of the other HIV-negative PEL cases suggests that age-related immunosenescence may be involved in pathogenesis [22,23].

In this study, concurrent EBV infection was seen in about two thirds of all patients, including about 75% of HIV-positive patients and 38% of HIV-negative patients, indicating a potential synergistic interaction between EBV and HHV8 in the pathogenesis of the disease, especially in HIV-positive patients. EBV encodes six nuclear transformation-associated proteins, EBNA 1–6, and immortalizes the infected B cells [24,25]. In patients with HIV infection, certain types of aggressive B-cell lymphomas, such as PEL and plasma-blastic lymphoma, represent EBV-transformed B-cells growing in an uncontrolled manner in an environment devoid of an appropriate immunoregulation [26,27]. 

Both PEL and Kaposi sarcoma are universally associated with HHV8 infection [19]. In the current study, about 30% of HIV-positive patients with PEL had Kaposi sarcoma. HIV-negative PEL patients did not develop Kaposi sarcoma. In the era before HAART therapy, Kaposi sarcoma was seen in up to 30% of HIV-positive patients [28]. In the current HAART therapy era, the incidence of Kaposi sarcoma has been markedly reduced [29]. The finding in this study that about 30% of HIV-positive PEL patients developed Kaposi sarcoma supports the experience in the literature [28,29]. Immune reconstitution using HAART therapy, although helpful, is not completely effective in the treatment of Kaposi sarcoma. HHV8 normally cannot be completely eradicated by using HARRT, and therefore, after complete remission of Kaposi sarcoma, there remains a risk of developing recurrent Kaposi sarcoma as well as PEL [30,31]. 

The diagnosis of PEL, although rare, should be considered in patients with HIV infection or other immunocompromised statuses. It is important to be aware of the morphological spectrum of PEL including plasmablastic, immunoblastic, and anaplastic features to avoid misinterpretation of these cases. Based on our experience, a diagnostic algorithm for the workup of large cell lymphoma with plasmablastic features is proposed in Figure 6. Both HHV8 immunostain and in situ hybridization for EBER should be performed in all cases of large cells lymphoma with plasmablastic morphology, especially in HIV-positive patients [26,32]. The minimal diagnostic panel for these lymphomas should include CD3, CD20, PAX5, CD138, MUM-1, ALK-1, HHV8, and EBER in situ hybridization. MUM-1 stain is usually positive in these large cell lymphomas. Plasmablastic lymphoma often has a similar clinical context (e.g., HIV-positive), cytomorphology, and immunophenotype (e.g., CD138-positive) as does PEL, except plasmablastic lymphoma is negative for HHV8. In addition, a potential pitfall is that PEL can be confused with T-cell lymphoma due to frequent expression of CD30 and occasional aberrant expression of CD3 in PEL (73.3% and 21.5% of cases respectively in this study). Therefore, additional T-cell markers and cytotoxic markers should be included to exclude anaplastic large cell lymphoma and peripheral T-cell lymphoma. Notably, expression of commonly utilized B-cell markers (such as CD20, CD79a, and PAX5) is often absent in PEL. Therefore, strong expression of CD20 and PAX5 argues against the diagnosis of PEL. PEL showed an extra copy of *MYC* gene but no evidence of *MYC* rearrangement in this study. *MYC* rearrangement by FISH analysis might be considered as a part of the diagnostic work-up, as *MYC* rearrangement is most often associated with plasmablastic lymphoma and anaplastic plasmacytoma, but absent in ALK-positive large B-cell lymphoma and PEL [33]. 

PEL is an aggressive B cell lymphoma with a very poor prognosis, even with aggressive chemotherapy [12]. Thus far, there is no consensus or standard therapeutic regimen recommended for PEL. In this case series, the EPOCH regimen was used in most patients, and the CHOP regimen was used in some patients. There was no significant difference in the outcome of patients treated with these regimens in a limited number of patients. The lack of clinical follow-up in a number of patients and the heterogeneity of therapies are limitations of this study in the assessment of prognosis. 

In this study patients with PEL who were HIV-negative had a median overall survival of 6.8 months. In contrast, HIV-positive patients seemed to have a better survival although not significantly different. The overall survival in our case series where HAART therapy was administrated in HIV patients was much better than the median OS of six months in previous studies [12,13,34]. Our cohort included one HIV-positive patient with extracavitary-only PEL that we recently encountered shown in Figure 3. The patient received HAART therapy and his positron emission tomography-computed tomography scan performed prior to chemotherapy showed complete resolution of lymphadenopathy. This result is similar to two previous case reports of HIV-positive patients with PEL who achieved remission after HAART therapy [35,36]. The pathogenesis of PEL might be via the production of HIV proteins and cellular cytokines in the HIV-positive patients [13,15]. HAART therapy can be effective in immune reconstitution and plays an indirect inhibitory effect on the pathogenesis of PEL. Therefore, it seems reasonable to propose that optimization and monitoring of HIV control and immune function in HIV-positive patients could be of benefit in the patients with HIV-associated PEL. Furthermore, HAART therapy might have a synergic effect with chemotherapy in HIV-associated PEL patients. Therefore, the combination of HAART and chemotherapy regimens should be considered in the HIV-positive patients.

Due to its rarity, our current cohort of 70 PEL cases is considered the largest to date; however, it still has statistical limitation due to its sample size and the lack of complete information in many of the patients. For example, comparisons of three subgroups of patients with different presentations (effusion only, extracavitary only, and both) were not performed because there were only eight patients in the subgroup with both effusion and extracavitary disease in the study. Furthermore, the pathological data in most of these eight patients were incomplete. To avoid a low statistical power because of small sample size, we did not perform statistics testing to compare this subgroup with the other two subgroups. Similarly, although the median OS of HIV-positive patients appeared different from that of HIV-negative patients (42.5 and 6.8 months, respectively), there was no statistically significant difference due to the low number of HIV-negative patients (*n* = 8). This might result in a false negative conclusion. A higher power study with more cases of this rare entity is warranted to further investigate the differences between PEL subgroups.

## 5. Conclusions

In summary, PEL is an aggressive lymphoma overwhelmingly associated with male gender and HIV infection. Cases of PEL show variable morphologic features and a plasmablastic immunophenotype. In this study, we also showed some differences in clinicopathologic features and immunophenotype between effusion-only versus extracavitary cases of PEL. However, neither HIV status nor disease presentation predicted clinical outcome in this cohort. Despite aggressive therapy regimens, patients with PEL have a poor prognosis.

## Figures and Tables

**Figure 1 cancers-13-00878-f001:**
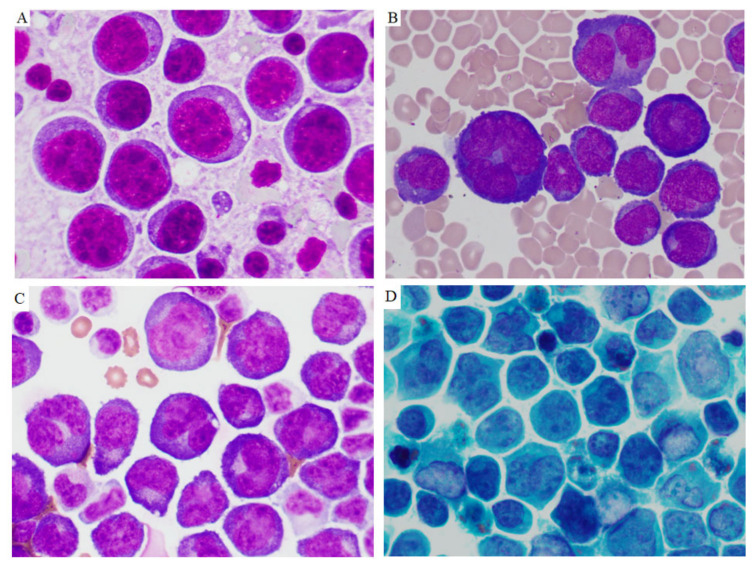
Cytomorphology of primary effusion lymphoma. The lymphoma cells from cytospins of effusions were large and had variable appearances, ranging from immunoblastic, plasmablastic to anaplastic morphology. Panels (**A**,**B**) were from two HIV-positive patients. Panels (**C**,**D**) were from one HIV-negative patient. (Panel (**A**–**C**) Diff-Quick stain. Panel (**D**) Papanicolaou stain. Original magnifications ×1000).

**Figure 2 cancers-13-00878-f002:**
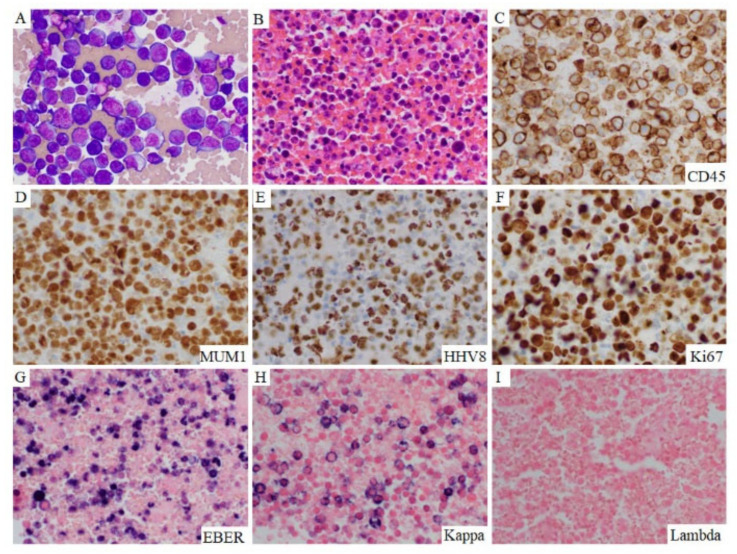
Morphology and phenotype of effusion-only PEL. This 65-year-old HIV-positive man who had excellent adherence to HAART therapy presented with a large pleural effusion. Morphologic examination of the pleural effusion showed abundant pleomorphic to anaplastic discohesive lymphoma cells (**A**), Diff-Quick stain. (**B**). H&E stained cell block). The lymphoma cells were positive for CD45 (**C**), MUM1 (**D**), HHV8 (**E**), Ki-67 (**F**), EBER (**G**), and monotypic Kappa by in situ hybridization (**H**,**I**). Original magnification ×500.

**Figure 3 cancers-13-00878-f003:**
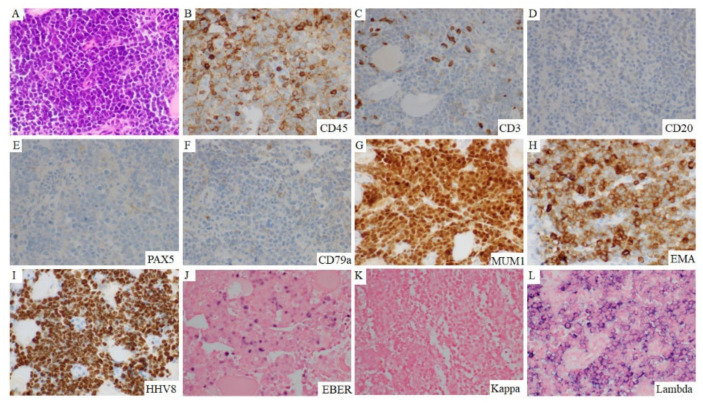
Morphology and phenotype of extracavitary-only PEL. This 31-year-old man who had poorly controlled HIV infection due to incompliance with HAART therapy presented with recent *pneumocystis jiriveci* infection and supraclavicular lymphadenopathy. A neck lymph node biopsy showed diffuse proliferation of large lymphoma cells (**A**). The lymphoma cells were positive for CD45 (**B**), MUM 1 (**G**), EMA (**H**), HHV8 (**I**), EBER (**J**), and monotypic lambda (**K**,**L**), and negative for CD3 (**C**), CD20 (**D**), PAX5 (**E**), and CD79a (**F**). Original magnification ×400.

**Figure 4 cancers-13-00878-f004:**
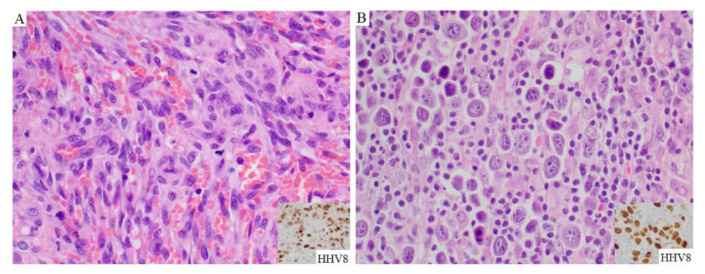
Histopathologic features of Kaposi sarcoma and PEL. The inguinal lymph node showed partially effaced nodal architecture by fascicles of spindled cells with extravasated red blood cells (**A**), hematoxylin-eosin, original magnification ×400). The spindle cells were positive for HHV8 stain (inset). In the cervical extracavitary mass of the same patient, the large lymphoma cells were distributed in a sinusoidal pattern (**B**), hematoxylin-eosin, original magnification ×400), and positive for HHV8 stain (inset).

**Figure 5 cancers-13-00878-f005:**
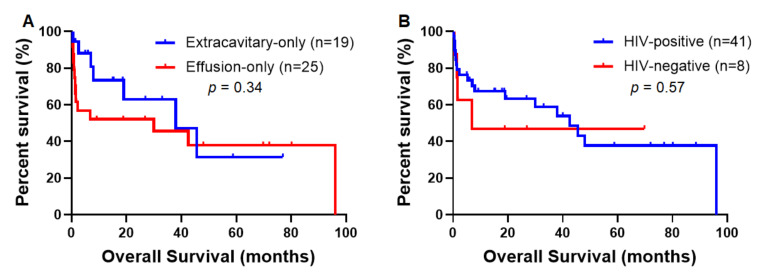
Prognostic impact of extracavitary presentation and HIV status. (**A**) Survival comparison between patients with effusion-only PEL and patients with extracavitary-only PEL. (**B**) Survival comparison between patients with HIV infection and patients without HIV infection.

**Figure 6 cancers-13-00878-f006:**
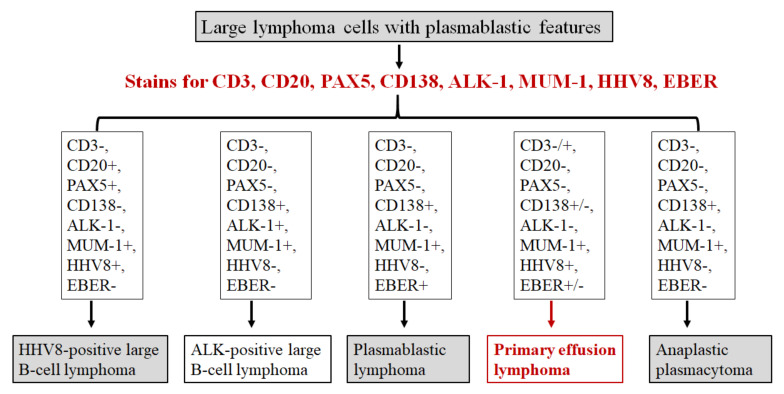
Diagnostic algorithm for large cell lymphoma with plasmablastic features. The minimal diagnostic panel for workup should include CD3, CD20, PAX5, CD138, MUM1, ALK-1, HHV8, and EBER. If desirable, a panel of T cell and cytotoxic markers is reasonable to exclude anaplastic large cell lymphoma. HHV8-positive large B-cell lymphoma might be considered when there is strong expression of CD20 and other B cell markers and HHV8 immunoreactivity in the tumor cells. ALK-1 expression is a key stain for ALK-positive large B-cell lymphoma.

**Table 1 cancers-13-00878-t001:** Clinical and laboratory characteristics of 70 patients with PEL.

Parameters	Results
Age, median (range)	46 (26–91)
Gender, N (%) Male Female	67 (95.7%) 3 (4.3%)
Immune status, N (%) HIV-positive ^1^ HIV-negative ^2^ Unknown	56 (80%) 8 (11.4%) 6 (8.6%)
CBC, N (%) Anemia Leukocytosis Leukopenia Thrombocytopenia	25/30 (83.3%) 4/30 (13.3%) 5/30 (16.7%) 14/29 (48.2%)
Disease presentation	
Effusion only, N (%) ^3^ Pleural Pericardial Peritoneal	35/70 (50.0%) 24/35 (68.6%) 8/35 (22.9%) 7/35 (20.0%)
Extracavitary only, N (%) ^4^ Lymph node Gastrointestinal tract Perirenal Skin/buccal Pelvic Central nerve system Atrium	27/70 (38.6%) 16/27 (59.3%) 8/27 (25.9%) 2/27 (7.5%) 2/27 (7.5%) 1/27 (3.7%) 1/27 (3.7%) 1/27 (3.7%)
Effusion and extracavitary, N (%) ^5^	8/70 (11.4%)
Treatment, N (%) EPOCH CHOP Other regimens ^6^	13/26 (50.0%) 8/26 (30.8%) 5/26 (19.2%)
Outcome Follow-up months, median (range) Medium OS (months) Deceased, N (%)	40.0 (0–96) 42.5 26/52 (50%)

Abbreviations: CBC, complete blood count; CHOP, cyclophosphamide, doxorubicin, vincristine, and prednisone; EPOCH, etoposide, doxorubicin, cyclophosphamide, vincristine, and prednisone; HIV, human immunodeficiency virus; OS, overall survival; PEL, primary effusion lymphoma. ^1^ Include 4 HBV-positive and 5 HCV-positive patients. ^2^ Include two patients with a history of heart transplant. ^3^ Include two patients with both pleural and pericardial effusions and two patients with both pleural and peritoneal effusions. ^4^ Include four patients with multiple masses (three with gastrointestinal tract and nodal; one with CNS and nodal). ^5^ Both effusion (four pleural, three peritoneal, and one pericardial) and extracavitary masses (three nodal, one duodenal, one perirenal, one inguinal and skin, one atrial, and one pleural). ^6^ Other chemotherapy regimens include two with doxil/paclitaxel, one with rituximab, ifofamide, carboplastin and etoposide, one with cyclophosphamide, vincristine, and prednisone, and one with Bendamustine.

**Table 2 cancers-13-00878-t002:** Phenotypic difference between effusion-only versus extracavitary-only PEL.

Markers	Overall	Effusion Only (*n* = 35)	Extracavitary Only (*n* = 27)	Both Effusion and Extracavitary (*n* = 8)	*p*-Value *
CD3	14/65 (21.5%)	4/30 (13.3%)	9/27 (33.3%)	1/8 (12.5%)	0.07
CD20	5/66 (7.6%)	1/32 (3.1%)	1/27 (3.7%)	3/7 (42.9%)	0.90
CD30	44/60 (73.3%)	22/27 (81.5%)	15/25 (60.0%)	7/8 (87.5%)	0.09
CD38	32/34 (94.1%)	19/20 (95.0%)	10/11 (90.9%)	3/3 (100%)	0.89
CD43	8/14 (57.1%)	4/7 (57.1%)	2/5 (40.0%)	2/2 (100%)	0.56
CD45	50/59 (84.7%)	25/26 (96.2%)	18/26 (69.2%)	7/7 (100%)	0.01
CD79a	5/37 (13.5%)	1/8 (12.5%)	4/25 (16.0%)	0/4 (0%)	0.81
CD138	30/58 (51.7%)	12/26 (46.1%)	15/25 (60.0%)	3/7 (42.9%)	0.32
ALK-1	0/34 (0%)	0/8 (0%)	0/23 (0%)	0/3 (0%)	NA
EMA	16/28 (57.1%)	10/10 (100%)	4/15(26.7%)	2/3 (66.7%)	0.0005 ***
HHV8	70/70 (100%)	35/35 (100%)	27/27 (100%)	8/8 (100%)	NA
PAX5 **	1/26 (3.8%)	1/9 (11.1%)	0/15 (0%)	0/2 (0%)	0.38 ***
MUM1	31/36 (86.1%)	14/14 (100.0%)	15/20 (72.2%)	2/2 (100.0%)	0.06 ***
Ki67 ≥ 70%	25/26 (96.1%)	6/6 (100%)	15/15 (100%)	4/5 (80.0%)	NA
EBER	39/60 (65.0%)	14/27 (51.9%)	20/26 (76.9%)	5/7 (71.4%)	0.06
K/L restriction	15/37 (40.5%)	6/20 (30.0%)	6/11 (54.5%)	3/6 (50%)	0.18

* *p*-value indicated the comparison of phenotypic parameters between effusion only and extracavitary only. ** Rare and weak staining positivity. *** By Fisher’s exact test. NA, statistics calculation cannot be performed.

**Table 3 cancers-13-00878-t003:** Clinical and laboratory difference between patients with effusion-only versus extracavitary-only PEL.

Parameters	Effusion Only (*n* = 35)	Extracavitary Only (*n* = 27)	Both Effusion and Extracavitary (*n* = 8)	*p*-Value *
Age (years)	52 (32–91)	42 (26–77)	43 (36–50)	0.001
Gender	33 M; 2 F	26 M; 1 F	8 M; 0 F	0.71
HIV-positive, N (%)	24/35 (68.6%)	24/27 (88.9%)	8/8 (100%)	0.06
Kaposi sarcoma, N (%)	9/19 (47.0%)	7/10 (70%)	3/5 (60%)	0.24
Elevated LDH, N (%)	9/14 (64.3%)	4/6 (66.7%)	3/5 (60%)	0.92
CBC, N (%) Anemia Thrombocytopenia Leukopenia	16/18 (88.9%) 7/17 (41.2%) 4/18 (22.2%)	6/9 (66.7%) 4/9 (44.4%) 1/9 (11.1%)	3/3 (100%) 3/3 (100%) 0/3 (0%)	0.16 0.87 0.48
Bone marrow involvement, N (%)	0/10 (0%)	0/9 (0%)	0/4 (0%)	NA
Gain of MYC/8q by FISH, N (%)	3/3 (100%)	1/2 (50.0%)	NA	0.40 **
Treatment, N (%) CHOP EPOCH	4/12 (33.3%) 4/12 (33.3%)	2/9 (22.2%) 6/9 (66.7%)	2/5 (40.0%) 3/5 (60.0%)	0.58 0.13
Outcome F/U time, median (range) (months) Mortality rate, N (%) Median OS (months)	48.0 (0–96) 16/26 (61.5%) 30	26.9 (0–76) 7/20 (35%) 38.0	54.1 (0–88) 3/6 (50%) 48	0.21 0.07 0.34

Abbreviations: CHOP, cyclophosphamide, doxorubicin, vincristine, and prednisone; EPOCH, etoposide, doxorubicin, cyclophosphamide, vincristine, and prednisone; FISH, fluorescence in situ hybridization; F/U, follow-up; N, number; NA, not applicable because Chi-square calculation cannot be performed; PEL, primary effusion lymphoma; OS, overall survival. * *p* values indicated the difference between effusion only vs. extracavitary only subgroups. ** Fisher’s Exact testing was used to compare the difference between effusion only vs. extracavitary only PELs.

**Table 4 cancers-13-00878-t004:** Clinical and laboratory difference between PEL patients with HIV versus without HIV infection.

Parameters	HIV-Positive (*n* = 56)	HIV-Negative (*n* = 8)	*p*-Value
Age (years)	45 (26–77)	75 (51–88)	0.005
Gender	55 M; 1 F	8 M; 0 F	1.00
Disease presentation, N (%)			
Effusion only	24/56 (42.9%)	7/8 (87.5%)	0.02 *
Extracavitary only	24/56 (42.9%)	1/8 (12.5%)	0.10
Both effusion and extracavitary	8/56 (14.3%)	0/8 (0%)	0.58 **
Kaposi sarcoma, N (%)	17/56 (30.4%)	0/8 (0%)	0.10 **
EBER-positive, N (%)	36/48 (75.0%)	3/8 (37.5%)	0.03 *
CBC, N (%) Anemia Thrombocytopenia Leukopenia	17/25 (68.0%) 13/24 (54.2%) 6/25 (24.0%)	6/6 (100%) 0/6 (0%) 0/6 (0%)	0.30 ** 0.02 ** 0.31 **
Bone marrow involvement, N (%)	0/18 (0%)	0/5 (0%)	NA
Gain of MYC/8q by FISH, N (%)	3/4 (75%)	1/1 (100.0%)	1.00
Treatment, N (%) CHOP EPOCH	7/21 (33.3%) 11/21 (52.4%)	1/5 (20.0%) 2/5 (40.0%)	0.56 0.62
Outcome F/U time, median (range) (months) Mortality rate, N (%) Median OS (months)	40.0 (0–70) 21/42 (50.0%) 42.5	26.9 (0–96) 4/8 (50.0%) 6.8	0.33 1.0 0.57

Abbreviations: CHOP, cyclophosphamide, doxorubicin, vincristine, and prednisone; EPOCH, etoposide, doxorubicin, cyclophosphamide, vincristine, and prednisone; FISH, fluorescence in situ hybridization; EBER, EBV-encoded small RNAs; F/U, follow-up; N, number; PEL, primary effusion lymphoma; OS, overall survival; NA, not applicable because statistics calculation cannot be performed. * *p* values indicate the difference between effusion only vs. extracavitary only subgroups. ** Fisher’s Exact testing was used to compare the difference between effusion only vs. extracavitary only PELs.

## Data Availability

Data is contained within the article.

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
