# Peer review of "Primary Effusion Lymphoma: A Clinicopathological Study of 70 Cases"

_cancers, 2021, doi:10.3390/cancers13040878_

Round 1
Reviewer 1 Report
This manuscript evaluates the clinical, pathological and molecular features of a selected population of 70 patients with primary effusion lymphoma (PEL), highlighting the most significant differences, also evaluating the presence or absence of HIV infection.
The manuscript is very well written; clear, precise, and easy to understand.
Minor points:
- In the results section, the clinical features, especially in the final part of the paragraph, are somewhat confusing. The authors could be more concise and use a differently structured table.
- The manuscript could benefit from the iconographic presentation of three patients: two PEL presenting with effusion-only, with and without a history of HIV infection and a third case with extra-cavitary mass and a history of HIV infection, in which the most significant differences (LCA, CD30 and EMA) could emerge.
- Finally, the authors should correct some writing errors (for example, in figure 6, PEL group, ALK should be completed).
Author Response
This manuscript evaluates the clinical, pathological and molecular features of a selected population of 70 patients with primary effusion lymphoma (PEL), highlighting the most significant differences, also evaluating the presence or absence of HIV infection.
The manuscript is very well written; clear, precise, and easy to understand.
Answer: We greatly appreciate the reviewer’s positive comment.
Minor points:
In the results section, the clinical features, especially in the final part of the paragraph, are somewhat confusing. The authors could be more concise and use a differently structured table.
Answer: per the reviewer’s suggestion, we restructured “Disease presentation” and removed “Bone marrow involvement” in Table 1 to make it more concise and easier to follow. In addition, we rephrased the sentence in the main text: page 4, lines 169-170 to make it easier to follow.
The manuscript could benefit from the iconographic presentation of three patients: two PEL presenting with effusion-only, with and without a history of HIV infection and a third case with extra-cavitary mass and a history of HIV infection, in which the most significant differences (LCA, CD30 and EMA) could emerge.
Answer: per the reviewer’s suggestions, we revised Figure 1 to include three cases of effusion-only PEL (the first two cases are HIV-positive and the other is HIV-negative). Figures 2-4 include HIV-positive cases.
Finally, the authors should correct some writing errors (for example, in figure 6, PEL group, ALK should be completed).
Answer: the writing errors in Figure 6 were corrected.

Reviewer 2 Report
In this manuscript, Hu et al describe the clinical, pathological and outcomes of 70 patients with primary effusion lymphoma diagnosed in 9 institutions over 15 years. They further compare effusion versus extracavitary tumors and HIV+ vs HIV-. Their findings are of interest and, despite the unavoidable limitations that accompany the study design, the rarity of this diagnosis makes higher quality evidence very hard to obtain. I find the reporting generally fair, clearly written and quite comprehensive. However, I do have a major issue with the manuscript, as well as minor comments and suggestions that I would like to know the authors’ idea based on the experience and knowledge this work will have brought them.
Major comments
1) I generally disagree with most of the outcome analysis. The outcome-centered conclusions are way too strong for this potentially biased and underpowered study. I think the clinicopathological findings and descriptive survival data are the core of the study and are relevant data for hematologists/oncologists and pathologist. I do not think the survival comparisons can be anywhere near the main takeaways from this study (first paragraph of the discussion mentions a lack of prognostic differences between effusion vs. extracavitary PEL). While I am aware that higher quality data may not be available, this does not change the fact that the comparisons are biased and vastly underpowered. Stating that there were no survival differences between patients treated with EPOCH and CHOP with 13 and 8 patients in each group, knowing that treatment was not randomly allocated but rather than a number of variables including age, performance status, changing treatment patterns during the 15 years of the study etc etc are likely to have played a role, is not only uninformative but can actually be misleading. Effusion-only vs. extracavitary PEL: survival is similar but mortality is different. While possible, this is odd and I find it an unlikely result if follow-up is similar and the study power adequate. HIV+ vs. HIV- cannot be compared after documenting the number of differences in patient characteristics. That a median survival of 42.5 months is not statistically different from one of 6.8 months seems also indicative of a very low power. I think the descriptive survival data is important but I would do away altogether with the comparisons.
Minor comments (general)
2) There is comparatively little specific data on the flow cytometric analysis of the samples. Based on its nature, flow cytometry seems particularly suited to help in the diagnosis of PEL. Yet the authors seem to find little role for it, particularly when proposing their algorithm. Could the authors comment? While I understand that the long inclusion period and the number of institutions might make complicated the analysis of flow cytometry data, I would be very interested in a more in-depth take on it. Most importantly, marker expression of PEL is given as FC/IHC. I think an analysis of the discrepancies, if any, would be very interesting. For instance, discordant results between the two are not rare, due to a number of factors, and I would like to know, if there were some discrepancies that occurred in a greater frequency that one would expect. I would like to know whether expression of CD3, which as the authors point out can be a source of diagnostic errors, was always seen in both. Patterns of expression of CD38 (how often very bright, at a level of plasma cells etc). The authors mention monotypic kappa or lambda only 40%. I suppose the remaining 60% lacked surface Ig expression (surely not polyclonal?). This should be stated.
3) Also related to flow cytometry, I understand not providing a detailed of reagents in this study, where flow cytometry is largely ancillary and which spans 15 years in 9 institutions. However, interpretive aspects should be provided, including how files were analyzed, gating strategy (perhaps with a representative figure), positivity thresholds, reference populations, controls etc. PEL samples are not always easy to interpret, given the aberrancies, the dim expressions etc. Therefore, the details are essential to understand the flow cytometry results and, if answers to minor comment 2 above are provided, understand those as well.
4) I wonder about the extent to which these results are applicable to other regions. This is conducted in 9 academic (I assume) institutions in specific areas of the US. Since this disorder is very associated with HIV, patterns of HIV infection are bound to impact the results. One could think that HIV patients in these 9 academic institutions are not broadly representative of the HIV community as a whole (treatment provision, follow-up etc). One could imagine that perhaps in areas where patients have no access to (high-quality) treatment there might be more high burden PEL (combined extracavitary and effusion)? For instance, Castillo et al (2018) found that only 57% of PEL from the national cancer database were HIV+. Conversely, the inclusion of institutions from a number of US states could make the results more representative of the US rather than one region specifically. Could the authors comment on how the population in their catchment areas could impact the results and whether they think this could challenge the external validity of their findings?
5) Relatedly, there is little data on HIV viral load, CD4 counts, antiretroviral therapy etc from the patient cohort. Indeed, HAART is mentioned several times in the discussion but no data is provided about HAART use in the results.
6)It could be interesting to have some staging data. Did patients get a CT scan? PET scan?
7)Recent reports mention an HHV8-negative variant of PEL (kaji et al, 2020). Have the authors found any in their institutions
8)CD45 findings are interesting? Have the authors found any data in previous reports that might suggest frequent lack of expression only in extracavitary PEL?
9)I found different p values from Fisher and chi-square tests than those provided in the tables, both using graphpad prism (as the authors) and other calculators. For instance, the p value for CD45 (table 2) I found was 0.024. The chi-square for EBER test 0.14, p value for mortality (table 3) was 0.2, table 4 effusion only in HIV+ vs HIV- was 0.02. Please re-check
10)Chi-square rules do not allow a cell with a 0 value but some p values are provided for three group comparisons (which must be obtained with chi-square as Fisher test cannot be used). See for instance anemia in table 3. Please explain how that was done. Again if chi-square is used, recheck the methods section
11)A big limitation I see is that a very large number of comparisons (baseline characteristics, antigen expression, HIV+ vs. HIV, effusion vs extracavitary etc) is conducted and any p value <0.05 is accepted as significant. This undoubtably results in false positive findings (Ioannidis, 2005) and overblown effect sizes (Ioannidis, 2008). This is a huge limitation but it goes unmentioned in the discussion. Please do
Minor comments (specific)
12)Introduction: HHV8 is not defined when the abbreviation is first used
13)Methods: is univariate survival analysis ~ as the logrank test? If so the latter may be more familiar for most readers and may be a better way to phrase this. If not, please explain this analysis and why the logrank test, which is the most commonly used, was not chosen
14)Results: HIV+ and HIV-negative. I suggest either HIV+ and HIV- or HIV-positive and HIV-negative for consistency
15)Results: The treatment category of “other” should be better described. At least know whether active vs. palliative treatment was given. Was rituximab given to CD20-pos (albeit dim) patients
16)Results: Chi-square is mentioned throughout the results section (table footnotes) but it is not in the methods. Check the methods for comprehensiveness
17)Results: Table 2 footnote. Pleural only likely means effusion only?
18)Discussion (page 11). The authors mention agreement between their findings (30% Kaposi sarcoma) and the literature but no reference is provided. Please do.
Author Response
In this manuscript, Hu et al describe the clinical, pathological and outcomes of 70 patients with primary effusion lymphoma diagnosed in 9 institutions over 15 years. They further compare effusion versus extracavitary tumors and HIV+ vs HIV-. Their findings are of interest and, despite the unavoidable limitations that accompany the study design, the rarity of this diagnosis makes higher quality evidence very hard to obtain. I find the reporting generally fair, clearly written and quite comprehensive. However, I do have a major issue with the manuscript, as well as minor comments and suggestions that I would like to know the authors’ idea based on the experience and knowledge this work will have brought them.
Major comments
1) I generally disagree with most of the outcome analysis. The outcome-centered conclusions are way too strong for this potentially biased and underpowered study. I think the clinicopathological findings and descriptive survival data are the core of the study and are relevant data for hematologists/oncologists and pathologist. I do not think the survival comparisons can be anywhere near the main takeaways from this study (first paragraph of the discussion mentions a lack of prognostic differences between effusion vs. extracavitary PEL). While I am aware that higher quality data may not be available, this does not change the fact that the comparisons are biased and vastly underpowered. Stating that there were no survival differences between patients treated with EPOCH and CHOP with 13 and 8 patients in each group, knowing that treatment was not randomly allocated but rather than a number of variables including age, performance status, changing treatment patterns during the 15 years of the study etc etc are likely to have played a role, is not only uninformative but can actually be misleading. Effusion-only vs. extracavitary PEL: survival is similar but mortality is different. While possible, this is odd and I find it an unlikely result if follow-up is similar and the study power adequate. HIV+ vs. HIV- cannot be compared after documenting the number of differences in patient characteristics. That a median survival of 42.5 months is not statistically different from one of 6.8 months seems also indicative of a very low power. I think the descriptive survival data is important but I would do away altogether with the comparisons.
Answer: We completely agree with the reviewer the limitation of the outcome analysis in this study due to small sample sizes. Per the reviewer’s suggestion, we made following revisions:
- We removed the outcome comparison between patients treated with EPOCH and CHOP given the lower number of patients and heterogeneity of treatment and other confounding factors.
- We descriptively stated the median OS of patients with effusion only and patients with extracavitary disease and removed words “no statistically significant difference” in main text: page 8, lines 245-247.
- We descriptively stated the median OS of HIV-positive and HIV-negative patients and removed words “no statistically significant difference” in main text: pages 8-9, lines 261-263.
- In the Discussion, we added two paragraphs to emphasize the limitation of statistical analysis in this study, main text: page 11, lines 341-344, 359-369.
Minor comments (general)
2) There is comparatively little specific data on the flow cytometric analysis of the samples. Based on its nature, flow cytometry seems particularly suited to help in the diagnosis of PEL. Yet the authors seem to find little role for it, particularly when proposing their algorithm. Could the authors comment? While I understand that the long inclusion period and the number of institutions might make complicated the analysis of flow cytometry data, I would be very interested in a more in-depth take on it. Most importantly, marker expression of PEL is given as FC/IHC. I think an analysis of the discrepancies, if any, would be very interesting. For instance, discordant results between the two are not rare, due to a number of factors, and I would like to know, if there were some discrepancies that occurred in a greater frequency that one would expect. I would like to know whether expression of CD3, which as the authors point out can be a source of diagnostic errors, was always seen in both. Patterns of expression of CD38 (how often very bright, at a level of plasma cells etc). The authors mention monotypic kappa or lambda only 40%. I suppose the remaining 60% lacked surface Ig expression (surely not polyclonal?). This should be stated.
Answer: flow cytometry information was available in 33 patients. Lymphoma cells were detected in 29/33 (87.9%) patients. CD38 expression was positive in 26/27 (96.3%) cases, including 1/26 with dim expression, and 2/26 with subset expression. CD38 expression assessed by FCS/IHC was added to table 2. Surface CD3 expression assessed by FCS was positive in 4/27 cases, and 2/2 (100%) was confirmed by CD3 immunostain. Regarding the immunoglobulin light chain expression in PEL, 7/37 (18.9%) cases had kappa restriction, 8/37 (21.6%) had lambda restriction, and the remaining case lacked surface light chain expression. The information is now provided in the main text: page 7, lines 216-217.
3) Also related to flow cytometry, I understand not providing a detailed of reagents in this study, where flow cytometry is largely ancillary and which spans 15 years in 9 institutions. However, interpretive aspects should be provided, including how files were analyzed, gating strategy (perhaps with a representative figure), positivity thresholds, reference populations, controls etc. PEL samples are not always easy to interpret, given the aberrancies, the dim expressions etc. Therefore, the details are essential to understand the flow cytometry results and, if answers to minor comment 2 above are provided, understand those as well.
Answer: flow cytometry was performed according standard protocol using antibody isotype controls and normal lymphocytes as references. Data were analyzed using FCS Express. Antigen expression was considered positive when at least 20% of lymphoma cells express the antigens. The information is now provided in the main text: page 3, lines 106-109.
4) I wonder about the extent to which these results are applicable to other regions. This is conducted in 9 academic (I assume) institutions in specific areas of the US. Since this disorder is very associated with HIV, patterns of HIV infection are bound to impact the results. One could think that HIV patients in these 9 academic institutions are not broadly representative of the HIV community as a whole (treatment provision, follow-up etc). One could imagine that perhaps in areas where patients have no access to (high-quality) treatment there might be more high burden PEL (combined extracavitary and effusion)? For instance, Castillo et al (2018) found that only 57% of PEL from the national cancer database were HIV+. Conversely, the inclusion of institutions from a number of US states could make the results more representative of the US rather than one region specifically. Could the authors comment on how the population in their catchment areas could impact the results and whether they think this could challenge the external validity of their findings?
Answer: the cohort of 70 cases in this study also included 9 cases published previously (AJSP 2012 Aug;36(8):1129-40). Overall, the 70 cases were collected from 18 institutions. We agree with the reviewer that our current study might be more representative than the national cancer database.
5) Relatedly, there is little data on HIV viral load, CD4 counts, antiretroviral therapy etc from the patient cohort. Indeed, HAART is mentioned several times in the discussion but no data is provided about HAART use in the results.
Answer: we thank the reviewer for pointing out the critical point. Additional information regarding HIV viral load, CD4 count, HAART therapy was included in the main manuscript: page 3, lines 138-143.
6) It could be interesting to have some staging data. Did patients get a CT scan? PET scan?
Answer: effusion-only PEL did not have extracavitary disease by definition. Of 35 cases with extracavitary disease by CT or PET scan, 4 patients had two masses and all other patients had one mass only.
7) Recent reports mention an HHV8-negative variant of PEL (kaji et al, 2020). Have the authors found any in their institutions.
Answer: rare cases of HHV8-negative effusion-based lymphomas were reported in literature and encountered in our practice. However, they are not considered as PEL in current WHO classification.
8) CD45 findings are interesting? Have the authors found any data in previous reports that might suggest frequent lack of expression only in extracavitary PEL?
Answer: the frequent lack of CD45 expression in extracavitary PEL was previously reported (AJSP 2012 Aug;36(8):1129-40) and documented in the current WHO classification. We include the information in the main text: page 10, line 284-285.
9) I found different p values from Fisher and chi-square tests than those provided in the tables, both using graphpad prism (as the authors) and other calculators. For instance, the p value for CD45 (table 2) I found was 0.024. The chi-square for EBER test 0.14, p value for mortality (table 3) was 0.2, table 4 effusion only in HIV+ vs HIV- was 0.02. Please re-check.
Answer: per the reviewer’s suggestion, we re-checked all p values and revised correspondingly in the revised manuscript.
10) Chi-square rules do not allow a cell with a 0 value but some p values are provided for three group comparisons (which must be obtained with chi-square as Fisher test cannot be used). See for instance anemia in table 3. Please explain how that was done. Again if chi-square is used, recheck the methods section.
Answer: three group comparisons were not performed in our current study. Chi-square and Fisher exact tests were used in the current study to compare the difference of parameters between effusion only and extracavitary only subtypes. Per the reviewer’s suggestions, we revised the description of statistics methods. “Chi-square calculations were used to compare the significance of difference of categorical variables. Fisher’s exact tests were performed when there was a cell with a value of “0”, and Chi-square tests were not allowed to perform for comparison”.
11) A big limitation I see is that a very large number of comparisons (baseline characteristics, antigen expression, HIV+ vs. HIV, effusion vs extracavitary etc) is conducted and any p value <0.05 is accepted as significant. This undoubtably results in false positive findings (Ioannidis, 2005) and overblown effect sizes (Ioannidis, 2008). This is a huge limitation but it goes unmentioned in the discussion. Please do.
Answer: we completely agree with the reviewer the limitation of the outcome analysis in this study. In the Discussion, we added two paragraphs to discuss the clinical and statistic limitations in the main text: page 11, lines 342-345 and lines 360-370.
Minor comments (specific)
12) Introduction: HHV8 is not defined when the abbreviation is first used.
Answer: HHV8 is now defined in the main text: page 2, lines 71.
13) Methods: is univariate survival analysis ~ as the logrank test? If so the latter may be more familiar for most readers and may be a better way to phrase this. If not, please explain this analysis and why the logrank test, which is the most commonly used, was not chosen.
Answer: log-rank test was used in our current study to compare the difference in survival of two independent groups. The information is included in page 3, line 129.
14) Results: HIV+ and HIV-negative. I suggest either HIV+ and HIV- or HIV-positive and HIV-negative for consistency.
Answer: per the reviewer’s suggestion, we revised HIV+, EBV+, HHV8+, and ALK+ to HIV-positive, EBV-positive, HHV8-positive, and ALK-positive, respectively, for consistency.
15) Results: The treatment category of “other” should be better described. At least know whether active vs. palliative treatment was given. Was rituximab given to CD20-pos (albeit dim) patients.
Answer: the treatment information in other patients is now included in the main text: page 8, lines 232-234, and in Table 1.
16) Results: Chi-square is mentioned throughout the results section (table footnotes) but it is not in the methods. Check the methods for comprehensiveness.
Answer: Chi-square test is now included in the Methods in the main text: page 3, line 127.
17) Results: Table 2 footnote. Pleural only likely means effusion only?
Answer: it was revised to effusion only
18) Discussion (page 11). The authors mention agreement between their findings (30% Kaposi sarcoma) and the literature but no reference is provided. Please do.
Answer: a new reference is now added in Discussion.

Reviewer 3 Report
The manuscript of Zhihong Hu and coll. titled “Primary Effusion Lymphoma: a Clinicopathological Study of 70 Cases” analyses a large series of Primary Effusion Lymphoma (PEL) highlighting differences in morphology, phenotype, clinical presentation and outcome between different type of PEL (which presents as effusion only or as extracavitary only or as both) and in different group of patients (HIV+ and HIV-). Although some differences between the groups are statistically significant, the clinical outcome and response to therapy are very similar in all groups underlining the aggressiveness of PEL and the poor prognosis of this type of lymphoma.
The only major criticism is the exclusion of PEL with both effusion and extracavitary presentation from analysis and discussion. For that reason, it is not clear as the p values of tables 2 and 3 have been obtained.
This group of cases looks more similar to effusion only PEL as regard of immunophenotype and more similar to extracavitary only PEL as regard of clinical parameters. Even if the significance of statistic may be affected by these differences it would be interesting (and will give more impact to the study) to analyze and discuss them.
In Table 2 data for CD38, CD43 and kappa/lambda are not reported. Moreover, the number of positive cases/total cases differs for CD3, CD79a and PAX5 from text to table.
In Table 2, is the single asterisk for CD20 correct?
At page 7 the reference to the Table 1 at the end of the sentence “There was no evidence of MYC rearrangement” is correct?
In the diagnostic algorithm proposed PAX5 in not reported in figure 6 and the utility of MUM1 and CD138 should be discussed.
Author Response
The manuscript of Zhihong Hu and coll. titled “Primary Effusion Lymphoma: a Clinicopathological Study of 70 Cases” analyses a large series of Primary Effusion Lymphoma (PEL) highlighting differences in morphology, phenotype, clinical presentation and outcome between different type of PEL (which presents as effusion only or as extracavitary only or as both) and in different group of patients (HIV+ and HIV-). Although some differences between the groups are statistically significant, the clinical outcome and response to therapy are very similar in all groups underlining the aggressiveness of PEL and the poor prognosis of this type of lymphoma.
The only major criticism is the exclusion of PEL with both effusion and extracavitary presentation from analysis and discussion. For that reason, it is not clear as the p values of tables 2 and 3 have been obtained.
This group of cases looks more similar to effusion only PEL as regard of immunophenotype and more similar to extracavitary only PEL as regard of clinical parameters. Even if the significance of statistic may be affected by these differences it would be interesting (and will give more impact to the study) to analyze and discuss them.
Answer: we thank the reviewer for raising the important point. Too few cases with both effusion and extravitary disease (n=8) makes it challenging to compare this group vs two other groups. Additionally, many parameters in Tables 2 and 3 from these 8 patients are lacking. Indeed <=5 patients had information in half of the parameters in these two tables. Nevertheless, we agree with the review and discuss the limitation in the main text: page 11, lines 359-365
In Table 2 data for CD38, CD43 and kappa/lambda are not reported. Moreover, the number of positive cases/total cases differs for CD3, CD79a and PAX5 from text to table.
Answer: we thank the reviewer for pointing out the mistakes. Stains for CD38, CD43, and Kappa/lambda are now reported in Table 2. The number of positive cases/total cases for CD3, CD79a, and Pax-5 are now corrected.
In Table 2, is the single asterisk for CD20 correct?
Answer: we now removed the asterisk.
At page 7 the reference to the Table 1 at the end of the sentence “There was no evidence of MYC rearrangement” is correct?
Answer: The reviewer is right. We removed the reference to the Table 1.
In the diagnostic algorithm proposed PAX5 in not reported in figure 6 and the utility of MUM1 and CD138 should be discussed.
Answer: we now include PAX5 in Figure 6 and revise the main text: page 10, lines 317-319 accordingly.

Round 2
Reviewer 2 Report
I thank the authors for undertaking the requested work. I have no further comments and will only go back to two minor aspects of previous comments that remain unsolved for the authors' consideration.
1) The authors removed survival comparisons on a suggestion that they were uninformative or even misleading. However, removal is incomplete:
Second to last sentence in the abstract.
p values leftover after removing specific comments (to the effect that no differences were seen): page 8 line 247, page 9 line 263, table 4 last row, figure 5
2) Differences between flow cytometry and immunohistochemistry, where both were available and could be compared?
Author Response
I thank the authors for undertaking the requested work. I have no further comments and will only go back to two minor aspects of previous comments that remain unsolved for the authors' consideration.
1) The authors removed survival comparisons on a suggestion that they were uninformative or even misleading. However, removal is incomplete:
Second to last sentence in the abstract.
Answer: per the reviewer’s suggestion, we revised correspondingly in page 2, lines 60-64.
p values leftover after removing specific comments (to the effect that no differences were seen): page 8 line 247, page 9 line 263, table 4 last row, figure 5
Answer: we completely agree with the reviewer on the limitation of statistical analysis when using small sample size. It is unavoidable when studying rare entities like PEL. Removing the p values would make study look less complete when presenting two groups. In the revised manuscript, we did de-emphasize the value of these “p” and emphasized the limitation of the statistical analysis. To maintain the integrity of the clinical study, we would like to keep the p values.
2) Differences between flow cytometry and immunohistochemistry, where both were available and could be compared?
Answer: CD3, CD20, CD30, CD45, CD138, kappa, and lambda were the markers performed by both flow cytometry and immunohistochemistry. However, these markers were not repeated by immunohistochemistry in many cases when flow cytometry was performed. These markers were performed by both flow cytometry and immunohistochemistry on 9 cases (CD3), 12 cases (CD20), 19 cases, 19 cases (CD30), 20 cases (CD45), 11 cases (CD138), and 4 cases (kappa and lambda). No discrepancy of marker expression was identified between flow cytometry and immunohistochemical stains on all cases.

This manuscript is a resubmission of an earlier submission. The following is a list of the peer review reports and author responses from that submission.